# Molecular Epidemiology of SARS-CoV-2 in Bangladesh

**DOI:** 10.3390/v17040517

**Published:** 2025-04-01

**Authors:** Abu Sayeed Mohammad Mahmud, Patiyan Andersson, Dieter Bulach, Sebastian Duchene, Anders Goncalves da Silva, Chantel Lin, Torsten Seemann, Benjamin P. Howden, Timothy P. Stinear, Tarannum Taznin, Md. Ahashan Habib, Shahina Akter, Tanjina Akhtar Banu, Md. Murshed Hasan Sarkar, Barna Goswami, Iffat Jahan, Md. Salim Khan

**Affiliations:** 1Bangladesh Council of Scientific and Industrial Research, Dr. Qudrat-E-Khuda Road, Dhaka 1205, Bangladesh; mohammas@ansto.gov.au (A.S.M.M.); murshedhasan-raj@bcsir.gov.bd (M.M.H.S.);; 2Microbiological Diagnostic Unit Public Health Laboratory, Department of Microbiology and Immunology, Peter Doherty Institute for Infection and Immunity, University of Melbourne, Melbourne, VIC 3000, Australia; patiyan.andersson@unimelb.edu.au (P.A.); dieter.bulach@unimelb.edu.au (D.B.);; 3Department of Microbiology and Immunology, Peter Doherty Institute for Infection and Immunity, University of Melbourne, Melbourne, VIC 3000, Australia; sebastian.duchene@unimelb.edu.au (S.D.); tstinear@unimelb.edu.au (T.P.S.); 4Centre for Pathogen Genomics, University of Melbourne, Melbourne, VIC 3000, Australia; 5Department of Microbiology, Faculty of Biological Science and Technology, Jashore University of Science and Technology, Jashore 7408, Bangladesh

**Keywords:** SARS-CoV-2, mutation, phylogeny, Bangladesh, spike protein, epidemiology

## Abstract

Mutation is one of the most important drivers of viral evolution and genome variability, allowing viruses to potentially evade host immune responses and develop drug resistance. In the context of COVID-19, local genomic surveillance of circulating virus populations is therefore critical. The goals of this study were to describe the distribution of different SARS-CoV-2 lineages, assess their genomic differences, and infer virus importation events in Bangladesh. We individually aligned 1965 SARS-CoV-2 genome sequences obtained between April 2020 and June 2021 to the Wuhan-1 sequence and used the resulting multiple sequence alignment as input to infer a maximum likelihood phylogenetic tree. Sequences were assigned to lineages as described by the hierarchical Pangolin nomenclature scheme. We built a phylogeographic model using the virus population genome sequence variation to infer the number of virus importation events. We observed thirty-four lineages and sub-lineages in Bangladesh, with B.1.1.25 and its sub-lineages D.* (979 sequences) dominating, as well as the Beta variant of concern (VOC) B.1.351 and its sub-lineages B.1.351.* (403 sequences). The earliest B.1.1.25/D.* lineages likely resulted from multiple introductions, some of which led to larger outbreak clusters. There were 570 missense mutations, 426 synonymous mutations, 18 frameshift mutations, 7 deletions, 2 insertions, 10 changes at start/stop codons, and 64 mutations in intergenic or untranslated regions. According to phylogeographic modeling, there were 31 importation events into Bangladesh (95% CI: 27–36). Like elsewhere, Bangladesh has experienced distinct waves of dominant lineages during the COVID-19 pandemic; this study focuses on the emergence and displacement of the first wave-dominated lineage, which contains mutations seen in several VOCs and may have had a transmission advantage over the extant lineages.

## 1. Introduction

The severe acute respiratory syndrome coronavirus 2 (SARS-CoV-2), the causative agent of COVID-19, was first reported in Wuhan, China, in December 2019 [1]. The outbreak of SARS-CoV-2 was declared a pandemic by the World Health Organization (WHO) on 11 March 2020. The first COVID-19 case in Bangladesh was reported on 8 March 2020; as of 30 June 2021, there were 888,406 confirmed cases (21.6 cases/100,000/week) and 14,172 deaths (case fatality rate of 1.6%) [2]. While all clades of SARS-CoV-2 were introduced to Bangladesh early in the pandemic, the B.1.1.25 lineage, also known as clade 20B, rapidly became the most prevalent [3,4].

Theoretically, SARS-CoV-2 containment depends on restricting the reproduction number, or *R*_0_, to less than 1.0 [5,6]. In practice, this proved difficult. Early on, it was clear that the spread of the virus was not contained by isolating affected individuals [5] due to the prevalence of asymptomatic or mildly symptomatic diseases (both can go undiagnosed during surveillance) and the infectiousness of the virus. Another concern was the possible pre-symptomatic transmission from infected individuals. Importantly, contact tracing studies have demonstrated that infectiousness peaks just before the onset of symptoms and that pre-symptomatic and moderately symptomatic individuals frequently transmit the virus [7], while completely asymptomatic individuals also transmit the virus [8].

The implications of these epidemiological characteristics for preventing the pandemic were catastrophic for a country like Bangladesh. Through the pandemic, the Bangladesh government and public health officials have adopted several strategies to slow local transmission of the virus. These have included lockdowns, limiting personal movement, contact tracing, extensive testing, mandatory use of masks, school shutdowns, limiting international arrivals, imposition of thermal scanners, and quarantine for international arrivals [9]. However, these strategies have not met with the same success in Bangladesh as they have in other locations [10]. Several factors have contributed to this outcome in Bangladesh, such as high population density (up to ~46,000/km^2^ in the capital, Dhaka). This high population density, coupled with low income, means that access to basic sanitation measures is not guaranteed, and social distancing is often not possible [9].

Lockdowns were used effectively to slow transmission in other countries. This measure could not be used extensively in Bangladesh due to the catastrophic impact of lockdowns would have on the finances of low-income families [11]. Finally, additional elements have contributed to the increased transmission and failure to control COVID-19 in Bangladesh, including poor access to medical care and testing facilities [2,9,12], poor communication and coordination among government bodies, and large gatherings during lockdown [9].

Thus, it is important to identify effective responses to COVID-19 in Bangladesh and measure their impact to ensure their efficacy. However, accurate measures of the impact of any public health policy are likely going to be hindered by poor estimates of the actual number of cases and, thus, the basic reproduction number *R*_0_ and other epidemiological parameters of interest [12]. The reasons for the difficulty in obtaining reliable estimates include the previously noted inadequate access to testing centers, which are frequently situated in cities and not in rural regions, and low testing rates; as of 30 June 2021, the WHO reported a testing rate of 105/100,000/week [2].

Despite the challenges, excellent efforts were made to sequence SARS-CoV-2 from as many cases as possible, with 2485 sequences deposited in GISAID on 30 June 2021. The genomic data enabled researchers to find novel, regularly mutating positions that correlate with clade-defining sites, which can help in monitoring the spread of viruses and their various clades. It can also be used to estimate independent measurements of *R*_0_, the influence of different public health approaches on transmission rates, and the extent of underreporting [13]. In this study, the whole genomes of 660 SARS-CoV-2 samples collected between April 2020 and June 2021 were sequenced, and 1305 sets of sequencing data from the public domain were analyzed to infer genomic variants, lineages, phylodynamic, and mutational patterns.

## 2. Materials and Methods

### 2.1. Setting and Data Sources

The Bangladesh Council of Scientific and Industrial Research (BCSIR) sequenced COVID-19-positive samples from public health screening clinics in the eight administrative districts of Bangladesh. Over this period, the total number of cases rose from ~10,000 to 760,000 [WHO weekly Coronavirus disease (COVID-19) Bangladesh situation report]. A representative sample set for sequencing was randomly selected to reflect the relative population proportions in each district [2022 census] and is presented in Table 1. The Human Research Ethics Committee at the National Institute of Laboratory Medicine and Referral Center (NILMRC) approved the whole-genome sequencing of SARS-CoV-2.

### 2.2. Nucleotide Extraction and Sequencing

Viral nucleic acid was extracted from nasopharyngeal specimens using the PureLink™ Viral RNA/DNA Mini Kit (Thermo Fisher Scientific, Waltham, MA, USA). cDNA was generated using random hexamer primed reverse transcription; specifically, 20 μL of RNA extract, 660 μM dNTPs, 5 × RT Improm II reaction buffer (Promega, Madison, WI, USA), 50 ng hexanucleotides, 1.5 mM MgCl2, 20 U RNasin^®^ Plus RNase Inhibitor (Promega), and 1U of ImProm-II™ Reverse Transcriptase (Promega). SARS-CoV-2 positive specimens were identified using the Novel Coronavirus (2019-nCoV) Nucleic Acid Diagnostic Kit (Sansure Biotech, Hunan, China detecting N-gene and ORF 1ab-gene). Each specimen’s virus load was determined using an RT-qPCR assay targeting a conserved region of the envelope gene. Sequencing-ready libraries were prepared using cDNA from the CoV sample (CoVOC43), the viral pool sample (ViralPool) with Nextera Flex for Enrichment (Illumina, San Diego, CA, USA, Catalog no. 20025524), and IDT for Illumina Nextera DNA UD Indexes (Illumina, Catalog no. 20027213). The total DNA input used for tagmentation was between 10 and 1000 ng, as recommended. After tagmentation and amplification, samples were enriched with the Respiratory Virus Oligos Panel (Illumina, Catalog no. 20042472). After enrichment, the prepared libraries were quantified, pooled, and loaded onto the MiniSeq™ System, producing data sets for each specimen comprising 76 base paired-end reads.

### 2.3. Bioinformatic Analysis for Generating Sequencing Data

FASTQ data sets were exported from the local run manager to BaseSpace Hub, Illumina. DRAGEN RNA Pathogen Detection V3.5.14 (BaseSpace) generated the consensus genome sequence for each sample, and consensus sequences were checked using the CZID platform [14]. The consensus sequence for each sample was uploaded in Genome Detective Virus Tools [15] to investigate the impact of mutations, both in relation to changes to protein-coding regions and encoded proteins. The consensus sequences were then compared to SARS-CoV-2 genomic data at the China National Center for Bioinformation (CNCB) [16] and Nextstrain.org. All genomic differences are with reference to the Wuhan-1 strain of SARS-CoV-2 (GenBank accession: MN908947.3). All sequences (*n* = 660) were uploaded to GISAID.

### 2.4. Nucleotide Substitution Analysis and Phylogeny

To complement the 660 sequences generated by BCSIR, we downloaded the remaining 1305 Bangladesh sequences (>28,000 bp) from GISAID for samples collected in the period from 1 April 2020 to 30 June 2021 for a total of 1965 sequences. We aligned each sequence to the Wuhan-1 reference genome sequence using *MAFFT* (v7.480—21 June 2021) [17] and then consolidated to a single multiple sequence alignment. The multiple sequence alignment was cleaned by replacing all non-ACGT bases with “-” using *goalign replace* (v0.3.4). We then used *goalign clean seqs* (v0.3.4) to remove all sequences with > 0.05 proportion of gap sites (“-”). Gappy sites and sites with no phylogenetic information were removed using a *clipkit* in the *epic-smart-gap* mode (v1.1.3 with a patch to work with BioPython v1.79) [18]. The alignment was then compressed by removing duplicate sequences with *goalign dedup* (v0.3.4) and by removing duplicate columns with *goalign compress* (v0.3.4). The sequence alignment was then used as input for building a maximum likelihood phylogenetic tree using *FastTree* (v2.1.10 with double precision [18] using the *-fastest* and *-nosupport* options). Multifurcations in the inferred tree were resolved using *gotree resolve* (v0.4.1). We estimated the branch lengths of the resolved tree from the previous step using the column weights obtained from *goalign dedup* and the alignment used in *FastTree* as input into *raxml-ng-evaluate* (v 1.0.2 released on 22 February 2021 [19] with the option *-blmin 0.0000000001* and assuming a GTR+G4 substitution model [20]). The final tree was then processed with a *gotree brlen round* to precision 10^−6^ and *gotree collapse length* to remove zero-length branches. Duplicate samples (previously removed) were re-inserted in the tree using *gotree repopulate* with duplicate labels being accessed through the deduplication list produced with *gotree dedup* above. Finally, we used *Newick utilities* v1.6 [21] to reorder (*nw_order*) and root the tree on the Wuhan-1 branch (*nw_root*). Phylogenetic trees were visualized and edited using FigTree ver. 1.4 (http://tree.bio.ed.ac.uk/software/figtree/; accessed on 1 June 2021).

### 2.5. Lineage Assignment

Sequences were assigned to lineages as described by the hierarchical Pangolin nomenclature scheme [22]. The following versions were used for each subcomponent of the software: pangolin (3.1.11), pangoLEARN (2021-08-21), scorpio (0.3.12), constellations (0.0.15), and designations (1.2.76).

### 2.6. Number of Importation Events

To infer the number of importation events, geographic locations (Bangladesh or abroad) were treated as discrete states in a phylogeographic model [23]. This model describes geographic movement along a phylogenetic tree as the result of a Markovian process, where Markov jumps from abroad to Bangladesh correspond to importation events (Figure 1). The model was set up in BEAST1.10 [24], but due to the size of the data set, a fixed phylogenetic time tree was used as in previous studies of SARS-CoV-2 phylogeographic [25], instead of a sequence alignment.

## 3. Results

### 3.1. Characterization of Samples

We analyzed 660 SARS-CoV-2 genomes from samples collected between April 2020 and June 2021 and sequenced by BCSIR (Table 1; sample metadata is presented in Appendix A). The age (median 40.5 years, range 0–95 years) and gender profile (Male, 67%; Female, 33%) of the patients reflect the profile of COVID-19 cases in Bangladesh. Sequenced samples from the Dhaka division (59.1%) are overrepresented in the dataset when compared to the population proportion profile (26%) but consistent with the proportion of COVID-19 cases (63.9% of cases).

### 3.2. Phylogenetic Analysis of SARS-CoV-2

We compared the 660 SARS-CoV-2 genomes generated by BCSIR together with 1305 sequences from Bangladesh available in GISAID sequenced by other institutions; an overview of the relationship between the sequences is shown in Figure 2 as a tree inferred from a multiple sequence alignment of the 1965 genome sequences. Thirty-four lineages and sub-lineages were observed, dominated by B.1.1.25 and its sub-lineages D.* (979 sequences) and the Beta variant of concern (VOC) B.1.351 and its sub-lineages B.1.351.* (403 sequences). The Alpha VOC B.1.1.7/Q.* (92 sequences), Gamma VOC P.1.* (1 sequence), Delta VOC B.1.617.2/AY.* (77 sequences), and B.1.36.* lineages (75 sequences) were also observed. The remaining lineages occurred at a frequency of 1 to 108 (median 2 sequences) and included sequences assigned to higher-level Pango lineages such as B, B.1, and B.1.1. The phylogenetic clade containing 1076 B.1.1.25 sequences include 97 sequences that were either unassigned or assigned to higher-level lineages, most likely due to missing data at lineage-defining sites.

Lineages occurred in waves, with B.1.1.25/D.* lineages observed between April 2020 to March 2021, followed by Alpha lineages from December 2020 to June 2021, partly overlapping with Beta lineages from November 2020 to June 2021, and with Delta lineages appearing in April 2021 (Figure 3 and Figure 4). The earliest sequences that were assigned as B.1.1.25/D.* lineages in GISAID are from five samples collected in the United Kingdom and Germany on 31 March 2020 and 1 April 2020 [26]. However, 45/49 B.1.1.25 sequences from samples collected in April 2020 were submitted from Bangladesh. The clade structure in the phylogenetic analysis indicates that the 979 sequences assigned as B.1.1.25/D.* in the dataset likely result from multiple introductions to Bangladesh, some of which subsequently led to large outbreak clusters. Following the introduction of the B.1.1.25 lineage in late March/early April, it rapidly became dominant (Figure 3). In the dataset analyzed, the proportion of B.1.1.25 rose from 42.5% in April to 64.5% and 74.1% in May and June and then remained between 72 and 86% through to January 2021. During this time, there was also a consistent presence of B.1.36.* lineages, constituting between 3 and 12% of the total from May to December 2020. The B.1.1.25 and B.1.36.* lineages were rapidly replaced by the introduction of the Alpha and Beta VOCs in December 2021 (Figure 4).

### 3.3. Genomic Variations in SARS-CoV-2

The B.1.1.25/D.* sequences observed in the Bangladeshi dataset all contained mutations Orf1a: I300F, Orf1b: P314L, S: D614G, N: R203K, N: G204R. Among the sequences in the B.1.1.25 clade, a further 570 non-synonymous (missense) mutations, 426 synonymous mutations, 18 frameshift mutations, 7 inframe deletions, 2 inframe insertions, 10 mutations affecting start/stop codons, and 64 mutations located in intergenic or untranslated regions were observed in the lineage Appendix A shows the most prevalent mutations present in the B.1.1.25 clade and spike mutations present in at least two samples. In addition to the canonical mutations for the lineage, other mutations that have been associated with changes in biological characteristics of the virus and presence in VOCs include S: P681R (*n* = 197), S: L452R (*n* = 10), S: E484K (*n* = 9), and S: L452Q (*n* = 2). Across the samples in B.1.1.25 clade, we observe nine deletions (two deletions in non-coding regions and seven resulting in in-frame deletions); most deletions were detected at a low frequency, and some may represent technical errors introduced during the sequencing processing. While we are hesitant to over-interpret the significance of these deletions, there was a notable presence in eleven samples of the del: HV69/70 deletion in the spike protein characteristic of the SARS-CoV-2 lineage in the UK [27].

### 3.4. Importation Dynamics

The analysis of the Markov jump revealed a median of 31 importation events into Bangladesh from any other country (95% credible interval: 27–36). Importantly, this may be an underestimate due to the sampling effort within Bangladesh and abroad being neither random nor exhaustive at that time. The model also provided an estimate of the importation rate, which had a median of 0.37 importations per lineage per year (95% credible interval: 0.05–1.30), an estimate that carries the same caveats as the number of importation events. These parameters indicate an average of about 45 genomes per importation event (median: 46.90; 95% credible interval: 40.39–53.85).

## 4. Discussion

Across the world, the COVID-19 pandemic consisted of successive waves of dominating SARS-CoV-2 lineages, including in Bangladesh. During the first wave in Bangladesh, from March to November 2020, the B.1.1.25 lineage was particularly successful. Understanding the genomic characteristics of such successful lineages, together with the epidemiological data, can help with the retrospective evaluation of public health responses.

The B.1.1.25 lineage and its sub-lineages D.2-D.5 are defined by the mutations Orf1a: I300F, Orf1b: P314L, S: D614G, N: R203K, and N: G204R. The Orf1b: P314L and S: D614G mutations are present in all five lineages considered variants of concern (VOC): Alpha, Beta, Gamma, Delta, and Omicron. The two mutations in the nucleocapsid gene, R203K and G204R, are also found in the Alpha, Gamma, and Omicron lineages. The Orf1a: I300F mutation is not seen in any of the VOC or variant of interest (VOI) lineages and outside the B.1.1.25 lineage, seemingly only extensively found in the B.1.1.315/AD.* lineage. The described effects of these mutations range from increased transmissibility to the appearance of only being lineage-associated. The genes encoding the spike and nucleocapsid proteins have an important role in the adaption and evolution of SARS-CoV-2. In other related viruses, these proteins have a key role in the initial interaction between the virus particle and a susceptible new host [28].

The Orf1a: I300F and Orf1b: P314L mutations have not been associated with a change in biological activity, with the former likely a lineage-specific mutation, and the latter is strong linkage-disequilibrium with the S: D614G mutation [29]. However, enhanced phenotypic characteristics have been described for the S: D614G, N: R203K, and N: G204R mutations. The mutation (S: D614G) within the spike protein close to the receptor binding domain emerged simultaneously in several geographical areas of the world in March 2020, rapidly became dominant, and is now ubiquitous in current SARS-CoV-2 lineages [30]. It is thought to be associated with a moderate advantage in infectivity and transmissibility [31]. It has been shown that retroviruses pseudotyped with SG614 can infect ACE2-expressing cells more effectively than the S: D614 [32]. The N: R203K and N: G204R mutations have been associated with increased infectivity of the virus, which is proposed to be mediated through increased effectiveness of RNA packaging. Experiments using viral pseudo particles showed that the N: R203K mutation was associated with a 10-fold increase in mRNA delivery and expression, and a reverse genetics model resulted in 50-fold increased viral titers [33]. Nucleocapsid protein can induce both cell-mediated and humoral immune responses and has possible utility in vaccine production [34]. The co-occurring mutations N: R203K and N: G204R have also shown increased infectivity in human lung cells and hamsters [35]. While the S: D614G mutation became fixed in the SARS-CoV-2 B-lineage early in the pandemic, the N: R203K and N: G204R mutations appear to have emerged independently multiple times in the history of SARS-CoV-2, including among several VOCs. The convergent evolution around these sites is indicative of the significant advantage of the alternate variant.

Among the B.1.1.25 sequences studied in this work, there were mutations associated with altered biological properties of the virus that occurred at minor frequencies, such as S: P681R, S: L452R, and S: E484K. The S:681R mutation was found in 95% of samples in the largest sub-clade within the B.1.1.25 clade, suggestive of an associated transmission advantage. The S: P681R has also been observed in the Alpha, Gamma, Delta, and Omicron VOCs. The mutation is located next to the furin-binding pocket and has been shown to enhance the cleavage of the full-length spike protein into its subunits, which are thought to improve viral cell entry [36]. The S: L452R mutation, which was also found in Delta and Omicron, has been associated with enhanced cleavage of the spike protein and increased ability to infect lung tissues of humanized ACE2 mice [37]. The S: E484K mutation has been a source of concern with changes in the residue associated with immune escape and convergent evolution observed in several lineages, including near universal presence in Beta and Gamma, and to a lesser extent, Alpha and Delta VOCs. Mutations of the S:484 residue, located in the receptor binding domain, are thought to increase the human ACE2 binding capacity of the spike protein. In experimental systems the S: E484K mutation reduces the neutralization of convalescent sera, including post-vaccination sera [36]. Trials have also indicated reduced susceptibility to some monoclonal antibodies for lineages carrying S: E484K. While most of these mutations did not have a profound effect in the B.1.1.25 population of Bangladesh, the emergence and subsequent success of the S: P681R in 20% of the B.1.1.25 samples shows the essential role that genomic surveillance can play in identifying changes in the epidemiology of the virus and changes in characteristics such as morbidity, mortality, and transmissibility [38].

The B.1.1.25 lineage showed prolonged dominance in Bangladesh in 2020. The initial wave of COVID-19 in Bangladesh was associated with a diversity of lineages, similar to other countries, but the B.1.1.25 lineage effectively outcompeted most other lineages following its introduction. This may indicate that the B.1.1.25 lineage had a transmission advantage compared with contemporary lineages, associated with mutations subsequently seen in several VOCs. However, epidemiological factors may have contributed to the rapid expansion of this lineage. The closure of the international border on 21 March 2020 meant a reduction in the introduction of new lineages into the country and a competitive advantage over lineages already present in the country. Similarly, the rapid dissemination of COVID-19 to regional areas associated with the mass migration events following the declaration of a prolonged National General Holiday and closures of workplaces likely also contributed to the dispersion of B.1.1.25 across the country and founder effects in regional areas [39]. Interestingly, B.1.1.25 was introduced into Australia via a traveler from Bangladesh and a breach in quarantine procedures was the cause of the significant second wave, mainly in the state of Victoria [40]. It is possible that an efficient transmission dynamic associated with the B.1.1.25 lineage contributed to the emergence and rapid spread of infections in Australia despite highly stringent public health restrictions and follow-up procedures at the time.

The B.1.1.25 lineage likely originated in Europe and was introduced to Bangladesh via travel. More than 90% of the B.1.1.25 sequences in GISAID from samples collected in April 2020 were submitted from Bangladesh. Although the earliest sequences assigned to this lineage were collected and submitted from the United Kingdom and Germany, given that the earliest date of B. 1.1.25 detection in Bangladesh was 31 March 2020, it is possible that this lineage originated in Bangladesh but went unnoticed due to inadequate surveillance. In the phylogenetic analysis of our dataset, there are 40 sequences at the base of the B.1.1.25 clade, making it difficult to distinguish whether the clade arose from single or multiple introductions. The available sequences only represent a small proportion of cases and would be affected by sequencing sampling strategies at the time, making it difficult to conclusively describe the emergence and international dispersal of the B.1.1.25 lineage. However, the importation analysis shows evidence of at least 31 introductions of the B.1.1.25 lineage to the country. Previous analyses of data from Bangladesh have suggested at least two introductory events of the B.1.1.25 lineage, but these phylogeographic analyses presented in this work allow a more detailed understanding of the genomic diversity and their geographic context.

In this analysis, a total of 2585 nucleotide mutation events were observed relative to the reference genome (MN908947.3), and these mutations occurred in 633 different positions in 25 different proteins. This finding suggests that mutational diversity was strikingly high in Bangladesh. However, this is possibly inflated as a result of effective efforts to sequence a diverse set of cases. The virus naturally acquires additional mutations in various areas as time passes. However, no unique lineages have been found in the various divisions of Bangladesh that would support a background of higher mutation.

The emergence of highly transmissible variants of SARS-CoV-2 elsewhere in the world underscores the importance of genomic monitoring. The strategy of obtaining a representative sample of the prevalent virus genomic type circulating in Bangladesh is important for the local management of disease and for understanding the virus’s global evolution during the pandemic. Continued genomic monitoring allows the detection of new variant incursions and, perhaps, the local evolution of variants of concern.

## 5. Conclusions

We present an overview of the brief history of SARS-CoV-2 in Bangladesh by using a representative sample of patients from throughout the country. We propose that SARS-CoV-2 was introduced into Bangladesh on several occasions. Although many introductions may have been overlooked since most cases have not been sequenced, one introduction event led to the dominant lineage in Bangladesh; this introduction is likely to have been via Europe, given the presence of characteristic mutations in samples from the dominant lineage. It is inevitable that as time progresses, the virus accumulates more independent mutations in different locations. However, at this stage, there is no evidence of significant differences in the dominant lineage in Bangladesh that may be an indicator of a significant local shift in the epidemiology of SARS-CoV-2 in Bangladesh. The sampling of the cases in this study may be sparse, and it is unlikely to have captured all the divergence that has occurred. However, this is a comprehensive and important baseline study that enables us to understand the epidemiology of SARS-CoV-2 in Bangladesh.

## Figures and Tables

**Figure 1 viruses-17-00517-f001:**
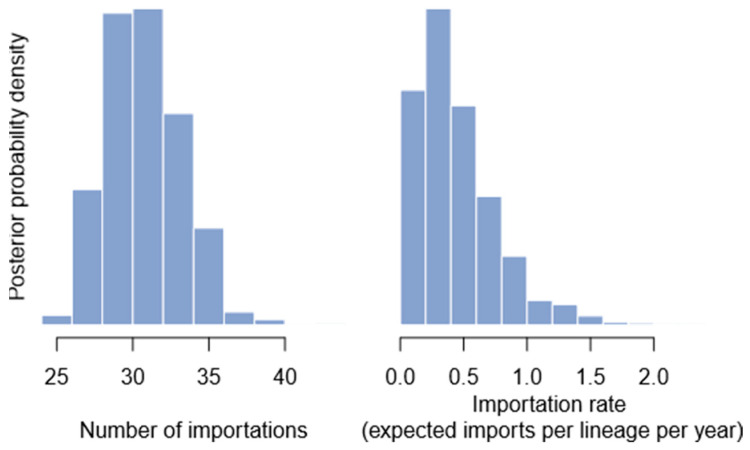
Histograms of posterior importation events and importation rate, as estimated using Markov jumps.

**Figure 2 viruses-17-00517-f002:**
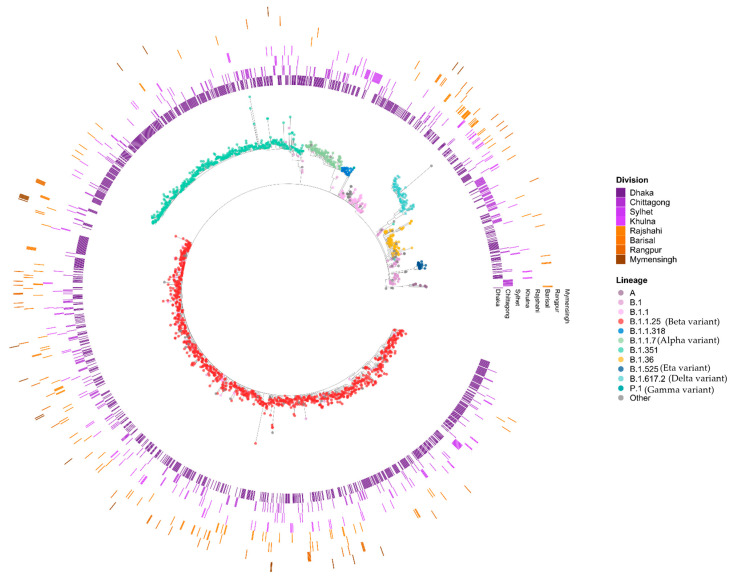
A phylogenetic tree showing the relationship between the SARS-CoV-2 genome sequences from each of the 1965 Bangladesh samples. Tips are colored by Pango lineage, with sub-lineages collapsed into the parental lineage, e.g., all D.* sub-lineages are counted in the B.1.1.25, all AY* sub-lineages in the B.1.617.2, all Q.* sub-lineages in the B.1.1.7, etc. Samples occurring at a frequency of fewer than 15 samples or that were unassigned were grouped into “Other”. The ring around the circumference of the tree shows the division from which each sample was collected.

**Figure 3 viruses-17-00517-f003:**
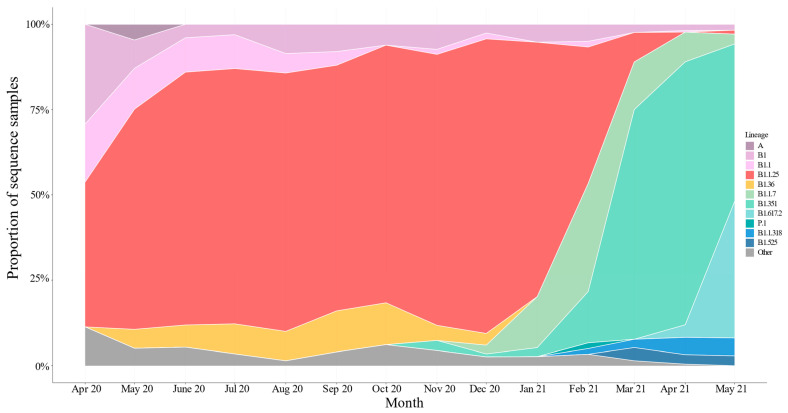
Proportional stacked area graph showing lineages over time in Bangladesh. Samples are collated by month and year of collection, and sub-lineages collapsed into the parental lineage, e.g., all D.* sub-lineages are counted in the B.1.1.25, all AY* sub-lineages in the B.1.617.2, all Q.* sub-lineages in B.1.1.7, etc. Samples occurring at a frequency of fewer than 15 samples or were unassigned were grouped into “Other”.

**Figure 4 viruses-17-00517-f004:**
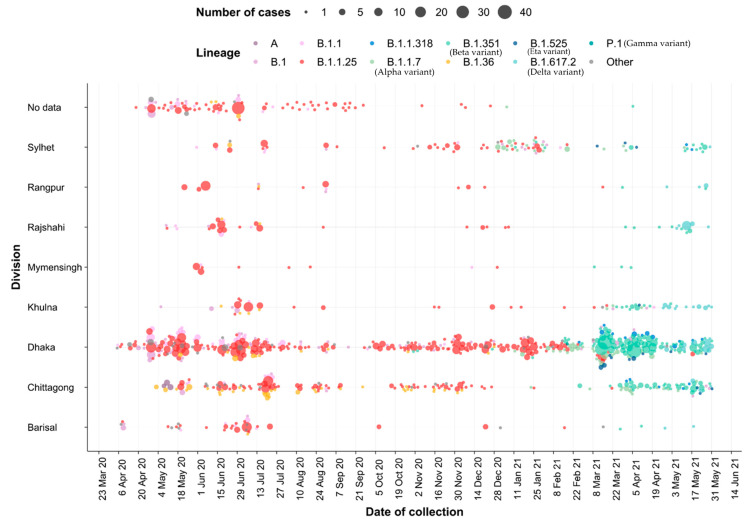
Dot plot graph showing lineages over time by division. The size of the dots is proportional to the number of samples observed for each lineage in each division for each day of collection. Dots are colored by Pango lineage, with sub-lineages collapsed into the parental lineage, e.g., all D.* sub-lineages are counted in the B.1.1.25, all AY* sub-lineages in the B.1.617.2, all Q.* sub-lineages in the B.1.1.7, etc. Samples occurring at a frequency of fewer than 15 samples or that were unassigned were grouped into “Other”.

**Table 1 viruses-17-00517-t001:** Geographic distribution (by Division) of patients from which virus genome sequence was obtained.

Division	* BCSIR Dataset	** Total Dataset	Population	COVID-19 Cases
Barishal	26 (3.9%)	70 (3.6%)	9,100,102 (5.5%)	2.4%
Chittagong	86 (13.0%)	295 (15.0%)	33,202,326 (20.1%)	13.4%
Dhaka	390 (59.1%)	1098 (55.9%)	44,215,107 (26.8%)	63.9%
Khulna	23 (3.5%)	91 (4.6%)	17,416,645 (10.5%)	6.1%
Mymensingh	14 (2.1%)	22 (1.1%)	12,225,498 (7.4%)	1.8%
Rajshahi	38 (5.8%)	78 (4.0%)	20,353,119 (12.3%)	5.6%
Rangpur	25 (3.8%)	44 (2.2%)	17,610,956 (10.7%)	3.4%
Sylhet	58 (8.8%)	116 (5.9%)	11,034,863 (6.7%)	3.4%
No data		151 (7.7%)		
Total	660		165,158,616 ^†^	327,349 ^

^†^ 2022 census; ^ 7 September 2020, WHO weekly Coronavirus disease (COVID-19) Bangladesh situation report. * BCSIR dataset: SARS-CoV-2 Sequenced by BCSIR. ** Total dataset: Gathered sequencing data from GISAID, incorporating the BCSIR dataset.

## Data Availability

The BCSIR-sequenced 660 SARS-CoV-2 genomes available at GISAID are presented in the table (see Appendix A). Moreover, 1365 SARS-CoV-2 sequences sequenced by other institutions in Bangladesh were collected from GISAID.

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
