# Peer review of "Molecular Epidemiology of SARS-CoV-2 in Bangladesh"

_viruses, 2025, doi:10.3390/v17040517_

Round 1

Reviewer 1 Report

Comments and Suggestions for Authors

The authors studied on the molecular epidemiology of SARS-CoV-2 infection in Bangladesh during 2020-21.Overall, the study design was well-conceived and clearly presented, although the use of older data (from two years ago) somewhat reduces its timeliness. Several concerns should be addressed prior to publication. 

Title: Since demographic data were incorporated, the title could be revised for clarity. I suggest the following: “Molecular Epidemiology of SARS-CoV-2 in Bangladesh during 2020-21. Additionally, please consider removing the subtitle for conciseness.

 Formatting: Adhere strictly to the “Instructions to the Authors” for manuscript formatting.

Replace the “Background” section heading with “Introduction” to align with common terminology.

Correct the font style in the final paragraph of the Introduction, as it currently deviates from the rest.

Ensure consistency in formatting, such as spacing and the use of italics, to avoid any misunderstandings regarding the quality of the manuscript. Such errors may lead to an underestimation of the manuscript's merit.

Author Response

Comment 1: Title: Since demographic data were incorporated, the title could be revised for clarity. I suggest the following: “Molecular Epidemiology of SARS-CoV-2 in Bangladesh during 2020-21. Additionally, please consider removing the subtitle for conciseness.

Response1: New title: Molecular Epidemiology of SARS-CoV-2 in Bangladesh during 2020-21.

Comment 2: Formatting: Adhere strictly to the “Instructions to the Authors” for manuscript formatting.

Response 2: Followed the instructions to the author.

Comment 3: Replace the “Background” section heading with “Introduction” to align with common terminology.

Response 3: Background replaced by introduction

Comments 4: Correct the font style in the final paragraph of the Introduction, as it currently deviates from the rest.

Response 4: Corrected to font "Palatino Linotype"

Comments 5:Ensure consistency in formatting, such as spacing and the use of italics, to avoid any misunderstandings regarding the quality of the manuscript. Such errors may lead to an underestimation of the manuscript's merit.

Response 5: Verified format and spacing

Reviewer 2 Report

Comments and Suggestions for Authors

The authors outlined the importation, transmission, and evolutionary history of SARS-CoV-2 in Bangladesh through sequencing and genomic epidemiological analysis of representative COVID-19 patient samples from various regions of Bangladesh. This study highlights the importance of genome sequencing in disease prevention and control, which is crucial for local disease management and understanding the global evolution of viruses during pandemics. Continuous genomic surveillance can detect the invasion of new variants and potentially identify the evolution of noteworthy variants locally.

Below are some comments and suggestions to help improve the clarity, depth, and overall impact of your manuscript.

1, The introduction section should minimize irrelevant information unrelated to the main content of the manuscript. For instance, what is the connection between the difficulty of COVID-19 prevention and control and the genomic epidemiology research discussed in this manuscript?

2, For the estimation of virus transmission routes, it is recommended to use SPREAD software. SPREAD software is primarily utilized in biogeography and phylogeography research, assisting scientists in comprehending the evolutionary history and geographical distribution patterns of biological populations.

3, Because this manuscript describes the genomic characteristics of SARS-CoV-2 in the early stages of the COVID-19 pandemic (April 2020 and June 2021), it is recommended to add a time limit to the title of the paper. Such as “Genomic Epidemiology of SARS-COV-2 in Bangladesh, April 2020 and June 2021”.

4, In Figures 1 and 2, and the main text, several familiar VOCs and VOIs should be labeled, such as B.1.1.7 (also labeled as Alpha variant), B.1.351 (also labeled as Beta variant), P.1 (also labeled as Gamma variant), B.1.617.2 (also labeled as Delta variant), and B.1.525 (also labeled as Eta variant).

Comments on the Quality of English Language

The English could be improved to more clearly express the research.

Author Response

Comment 1: The introduction section should minimize irrelevant information unrelated to the main content of the manuscript. For instance, what is the connection between the difficulty of COVID-19 prevention and control and the genomic epidemiology research discussed in this manuscript?

Response 1: The background clearly articulates the precise context of the COVID-19 pandemic. Two reviewers agreed with the introduction.

Comment 2: For the estimation of virus transmission routes, it is recommended to use SPREAD software. SPREAD software is primarily utilized in biogeography and phylogeography research, assisting scientists in comprehending the evolutionary history and geographical distribution patterns of biological populations.

Response 2: 

The software we have employed was published in Nature Microbiology; the following versions were used for each subcomponent of the software: pangolin (3.1.11), pangoLEARN (2021-08-24), scorpio (0.3.12), constellations (0.0.15), and designations (1.2.76).

Comment 3: Because this manuscript describes the genomic characteristics of SARS-CoV-2 in the early stages of the COVID-19 pandemic (April 2020 and June 2021), it is recommended to add a time limit to the title of the paper. Such as “Genomic Epidemiology of SARS-COV-2 in Bangladesh, April 2020 and June 2021”.

Response 3: Agreed, and change the title accordingly.

Comment 4: In Figures 1 and 2, and the main text, several familiar VOCs and VOIs should be labeled, such as B.1.1.7 (also labeled as Alpha variant), B.1.351 (also labeled as Beta variant), P.1 (also labeled as Gamma variant), B.1.617.2 (also labeled as Delta variant), and B.1.525 (also labeled as Eta variant).

Response 4: Agreed and labelled both figures accordingly.

Reviewer 3 Report

Comments and Suggestions for Authors

This study explores the genomic epidemiology of SARS-CoV-2 in Bangladesh, analyzing 1,965 genomes (comprising 660 newly sequenced genomes) collected between April 2020 and June 2021 to understand lineage diversity, mutations, and importation events. It identifies 34 lineages, with B.1.1.25/D.* and Beta (B.1.351) variants dominating, and infers 31 importation events, primarily from Europe. The analysis reveals 1,085 mutations, including key ones like S: D614G, which enhance infectivity, and S: P681R, which aids viral entry, contributing to the prolonged dominance of certain lineages. The findings highlight the critical role of genomic surveillance in tracking variant dynamics and adapting public health strategies, particularly in densely populated regions with limited healthcare resources. This work establishes a valuable baseline for understanding SARS-CoV-2's evolution and its public health impact in Bangladesh.

Review suggestions:

Table S1: Table S1, referenced in the manuscript, is not provided for review.

Clarification on Table 1: The term "Total dataset" in Table 1 requires clarification to ensure readers understand how it differs from the "BCSIR dataset" and whether it represents all available cases or a subset.

Timeliness of Data: The data collected in this study spans April 2020 to June 2021, which is somewhat outdated and may not capture the evolutionary dynamics or newer variants that have emerged post-2021.

Supplementary File 1: (Suppl. 1), mentioned in the manuscript, is not available for review.

Table 2: Table 2 is not included for review, hindering a comprehensive understanding of the most prevalent mutations and their significance in the context of the B.1.1.25 clade.

Undiscussed Figures: Figure 3 and the figure below it are neither cited nor discussed in the main text, making their relevance unclear.

Inaccuracy in Line 101: The claim in line 101 that the D614G mutation is located in the receptor binding domain (RBD) of the spike protein is inaccurate. D614G is within the spike protein but outside the RBD and should be corrected to avoid misleading readers.

Comments on the Quality of English Language

Fine.

Author Response

Table S1: Table S1, referenced in the manuscript, is not provided for review.

Comment 1: Clarification on Table 1: The term "Total dataset" in Table 1 requires clarification to ensure readers understand how it differs from the "BCSIR dataset" and whether it represents all available cases or a subset.

Response 1: Clarified and highlighted BCSIR dataset and Total dataset

Comment 2: Timeliness of Data: The data collected in this study spans April 2020 to June 2021, which is somewhat outdated and may not capture the evolutionary dynamics or newer variants that have emerged post-2021.

Response 2: The title of the articles has changed as per the reviewer suggestion.

Comment 3: Supplementary File 1: (Suppl. 1), mentioned in the manuscript, is not available for review.

Response 3: Uploaded suppl. 1 data

Comment 4: Table 2: Table 2 is not included for review, hindering a comprehensive understanding of the most prevalent mutations and their significance in the context of the B.1.1.25 clade.

Response 4: Table 2 is uploaded for review

Comment 5: Undiscussed Figures: Figure 3 and the figure below it are neither cited nor discussed in the main text, making their relevance unclear.

Response 5: Figure 3 is cited in the result section

Comment 6: Inaccuracy in Line 101: The claim in line 101 that the D614G mutation is located in the receptor binding domain (RBD) of the spike protein is inaccurate. D614G is within the spike protein but outside the RBD and should be corrected to avoid misleading readers.

Response 6: We have rewritten and highlighted this sentence.

Round 2

Reviewer 3 Report

Comments and Suggestions for Authors

The authors have addressed most of my concerns and revised the manuscript accordingly. The study presents valuable insights into the genomic epidemiology of SARS-CoV-2 in Bangladesh and is a meaningful contribution to the field. I recommend that the manuscript be accepted for publication in its current form.